# Proof2Hybrid: Automatic Mathematical Benchmark Synthesis for Proof-Centric Problems

## Abstract

Evaluating the mathematical capability of Large Language Models (LLMs) is a critical yet challenging frontier. Existing benchmarks fall short, particularly for proof-centric problems, as manual creation is unscalable and costly, leaving the true mathematical abilities of LLMs largely unassessed. To overcome these barriers, we propose Proof2Hybrid, the first fully automated framework that synthesizes high-quality, proof-centric benchmarks from natural language mathematical corpora. The key novelty of our solution is Proof2X, a roadmap of converting mathematical proofs into various kinds of questions that are easy to verify. Instructed by this roadmap, we propose a new type of hybrid-formatted questions, named "$m$-out-of-$n$ multiple judge questions", specifically designed to enable robust, automatic evaluation while being resilient to guessing and superficial pattern matching inherent in traditional formats. As a demonstration of our framework, we introduce AlgGeoTest, a benchmark for algebraic geometry—a frontier domain of modern mathematics—comprising 456 challenging items. Our extensive evaluations on state-of-the-art LLMs using AlgGeoTest reveal profound deficits in their comprehension of algebraic geometry, providing a more precise measure of their true mathematical capabilities. Our framework and benchmark pave the way for a new wave of in-depth research into the mathematical intelligence of AI systems. Our code and data are provided in the supplementary material.

## 1 Introduction

In recent years, large language models have developed at an astonishing pace (Anthropic, 2025b; Guo et al., 2025; Google, 2025; OpenAI, 2025a), far exceeding what most people imagined just a few years ago. Naturally, these models are now looking to tackle fields traditionally thought to be beyond artificial intelligence's reach, with cutting-edge mathematics being a prime example (Cai & Singh, 2025; Huang & Yang, 2025). A model's ability to understand mathematics serves as a strong indicator of its overall reasoning capability and intellectual depth. However, evaluating this understanding remains challenging in many areas of mathematics (Wang et al., 2025), mainly because we lack comprehensive benchmarks that cover a wide enough range of topics.

Mathematical problems generally fall into two categories: the number-centric problems which value numerical calculations and have a definite value as answer, and the proof-centric problems that value proofs and logical deductions and often has no definite value as answer. Number-centric problems—solutions of which being straightforward to verify—are primarily found in elementary areas such as high-school math or easy Olympiad math. This paper, however, focuses on proof-centric mathematical problems. These problems comprise the majority of cutting-edge mathematics and thus they are of great importance (Morris, 2020). LLMs' understanding of these problems relies heavily on the ability to construct valid proofs and evaluate the correctness of existing ones. Currently, there are few benchmarks focusing on proof-centric mathematical problems. Therefore, we aim to construct benchmarks to assess a model's understanding of proof-centric mathematical problems, providing insight for future pre-training and supervised fine-tuning.

Current benchmarks on the proof-centric problems are mainly constructed following two types of strategies: commissioning human experts to craft items (Glazer et al., 2024; Phan et al., 2025),

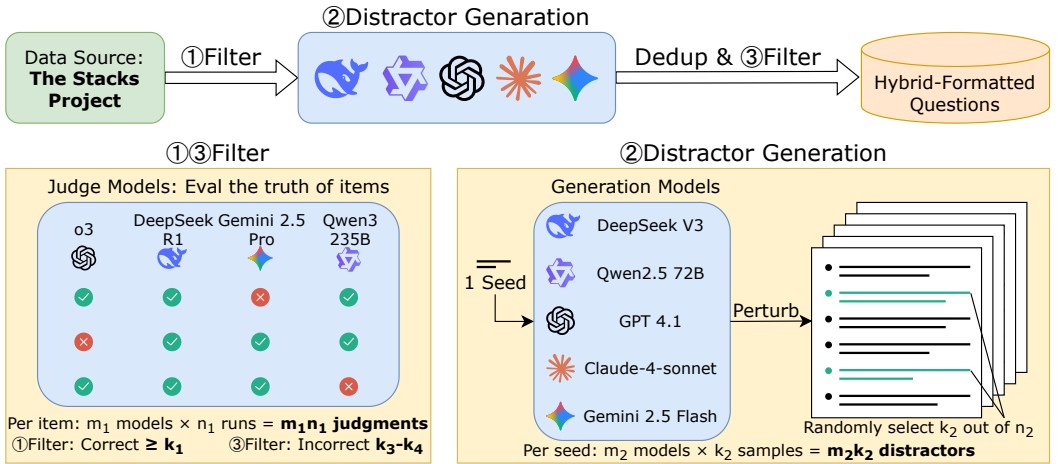

Figure 1: Example of the full workflow of Proof2Hybrid in our scenario of producing AlgGeoTest. At the framework's heart is a carefully orchestrated pipeline of powerful LLMs. First, a generation team of models crafts distractors by strategically altering keywords, conditions, or formulas in the original statements. Then, a separate judging team filters and refines these candidates, discarding any that are obviously wrong while retaining those that are subtly and deceptively flawed. This carefully engineered generation and verification process minimizes potential evaluation bias and guarantees the exceptional quality of the resulting questions. A sample problem is given in Appendix B, illustrating the whole pipeline.

and utilizing formal proof languages (de Moura & Ullrich, 2021; Paulin-Mohring, 2012; Nipkow et al., 2002) to synthesize tasks (Zheng et al., 2022; Tsoukalas et al., 2024). Both methods have apparent shortcomings in practice. The former yields high-quality benchmarks but is difficult to generalize across domains and practically impossible to scale, given the limited availability of expert mathematicians. The latter, although ostensibly domain-agnostic, still demands substantial manual effort for labeling and verification, making true horizontal scaling infeasible. Considering the lack of handy benchmarks in practice, we want to propose a fully-automated framework that can efficiently construct benchmarks focusing on proof-centric mathematical problems.

Our solution to this task is **Proof2Hybrid**, the first **fully automated** framework for proof-centric mathematical benchmark synthesis, which is also domain-agnostic, scalable and cost-effective. Benchmarks generated by our framework can rigorously evaluate LLMs' mathematical ability in certain mathematical domains, and also allow flexible adjustment of difficulty.

The key novelty of our solution is **Proof2X**: a newly proposed roadmap converting mathematical proofs—abundant in natural corpora—into various kinds of questions that are easy to verify. The "X" in Proof2X could be multiple-choice problems, true-or-false problems, blank-filling problems or any other kinds of problems whose answer can be verified automatically by a program.

Instructed by this roadmap, we propose a new type of hybrid-formatted questions, named "$m$-out-of-$n$ multiple judge question", which eliminates the disadvantages of solely using multiple choice questions or true-or-false questions. Aiming to make the target question type easily verified, a natural choice is using true-or-false questions to ask LLMs to judge the correctness of a proof. However, there were several defects with true-or-false questions. Firstly, the answer is relatively easy to guess, since random guessing achieves an expected accuracy of $\frac{1}{2}$. Secondly, the standards of "a correct proof" differ significantly among different models: There are models that do not allow the slightest ambiguity of expression, while there are also models that are very tolerant towards mistakes because they view these as clerical errors. Hence, we can not assume that the standard of every participant maintains consistency. Another straight forward solution is multiple choice questions, which means letting LLMs choose the most rigorous proof among multiple terms. This sort of question form incurs another problem: models might compare the given choices and try to guess the answer based on non-mathematical patterns. Therefore, we propose our hybrid question format, "$m$-out-of-$n$ multiple judge question". This question format gives LLMs $n$ items, each item comprise a mathematical proposition and its proof, which can be either correct or incorrect. We

Table 1: Pros and cons comparison between AlgGeoTest and other mathematical benchmarks. Meaning of Abbreviations: **PC**: proof-centric. **NL**: natural language. **LS**: large scale (more than 200 questions). **AUTO**: automatic (without human effort during the generation phase).

|  | PC | NL | LS | AUTO |
|---|---|---|---|---|
| MATH (Hendrycks et al., 2021) |  | ✓ | ✓ |  |
| AIME (, Maxwell-Jia) |  | ✓ |  |  |
| PutnamBench (Tsoukalas et al., 2024) | ✓ |  | ✓ |  |
| HLE-MATH (Phan et al., 2025) | ✓ | ✓ | ✓ |  |
| Omni-MATH (Gao et al., 2024) | ✓ | ✓ | ✓ |  |
| IMO (IMO-Board, 2025) | ✓ | ✓ |  |  |
| **AlgGeoTest** | ✓ | ✓ | ✓ | ✓ |

ensure that all items in a question stem from different mathematical propositions, and exactly $m$ out of the $n$ items are correct. We find that this question type effectively eliminates all drawbacks mentioned above.

To anchor Proof2Hybrid in a firm foundation, we instantiate it on the definitions and propositions of the open source Algebraic Geometry textbook and reference work "The Stacks project" (Stacks-Project-Authors, 2025). The result is **AlgGeoTest**, a benchmark designed to probe large language models' grasp of Algebraic Geometry—a frontier domain of modern mathematics that occupies a central position within the contemporary mathematical landscape. AlgGeoTest contains 456 items, each offering six true-or-false sub-questions: exactly two true subquestions and four carefully engineered distractors. Our evaluation results and audit outcomes safeguard the benchmark's quality and thus offer compelling evidence for the robustness of Proof2Hybrid. We also propose a perplexity-based evaluation protocol for our benchmark aimed to lighten LLMs' cognitive load, which is especially suitable for evaluating base models. Our code and data are provided in the supplementary material.

## 2 RELATED WORK

To comprehensively evaluate the mathematical ability of LLMs, various benchmarks have been created, including GSM8K, MATH, MMLU-Pro (Math) and AIME 2024 (Cobbe et al., 2021; Hendrycks et al., 2021; Wang et al., 2024; , Maxwell-Jia). The common point of these benchmarks is that all their questions have a definite answer consisting of numerical values or explicit formulae, thereby constraining their focus to relatively simple and straightforward math problems. However, proof-centric questions, which require rigorous logical deduction and structured problem solving, remain largely unaddressed by these benchmarks.

As LLMs advance, state-of-the-art LLMs have also reached performance saturation on the above benchmarks (Vendrow et al., 2025). As a result, harder mathematical benchmarks have been established in order to further boost up LLM's mathematical performance. Currently there are two mainstream approaches to construct more challenging math benchmarks. The first method is to extract questions from the most difficult math Olympics such as IMO (International Mathematics Olympiad, (IMO-Board, 2025)). For instance, Omni-MATH (Gao et al., 2024) proposes a comprehensive and challenging benchmark particularly designed to assess LLMs' mathematical reasoning at the Olympiad level, enabling a nuanced analysis of model performance across different levels of complexity. Similarly, OlymMATH (Sun et al., 2025) proposes a benchmark of Olympiad level mathematical questions spanning multiple mathematical domains, including algebra and geometry. However, this approach is significantly constrained by the scarcity of IMO-level math questions with well-explained solutions and precise, detailed answers. The second approach involves constructing challenging mathematical benchmarks manually curated by expert mathematicians. Notable examples of this method include FrontierMath (Glazer et al., 2024) and HLE-Math (Phan et al., 2025). However, this approach is inherently unsustainable at scale, as it relies on the limited availability of highly trained experts, making the scalable creation of such benchmarks impractical.

There are also benchmarks focusing on math proof questions. For example, miniF2F (Zheng et al., 2022) serves as an initial effort towards evaluating neural mathematical reasoning capabilities in formal environments, utilizing Lean (de Moura & Ullrich, 2021) to assess the performance of neural theorem provers. PutnamBench (Tsoukalas et al., 2024) enrich the dataset to more than 1,500 hand-crafted formalizations written primarily in Lean 4, Coq, and Isabelle (de Moura & Ullrich, 2021; Paulin-Mohring, 2012; Nipkow et al., 2002). Since the formalization of mathematical prob-

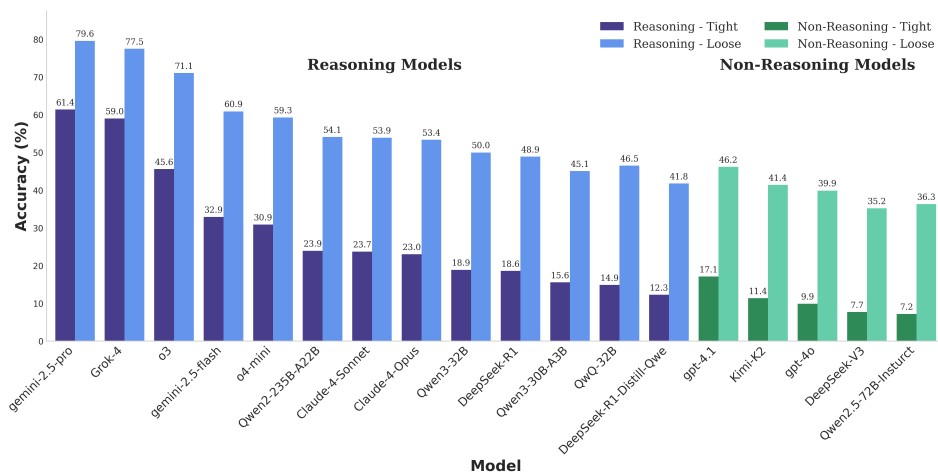

Figure 2: The evaluation results of AlgGeoTest on multiple LLMs. The evaluation results reveal that even the best-performing model to date achieves only a moderate score of around 60, with scores commonly lower than 20. This result underscores the rigorous and challenging nature of our benchmark. We also observe that reasoning models commonly outperform their non-reasoning counterparts, demonstrating that our benchmark effectively measures the reasoning capability of LLMs.

lems using formal proof languages such as Lean demands extensive human efforts, producing such benchmarks at scale remains costly and labor-intensive.

AlgGeoTest addresses these gaps by introducing an LLM-based methodology, named Proof2Hybrid, to automatically adapt proof questions into a hybrid multi-choice problem, where each choice represents a True or False statement. A comparison between AlgGeoTest and other mathematical benchmarks is presented in Table 1.

## 3 THE PROOF2HYBRID FRAMEWORK

In this section we present the full workflow of **Proof2Hybrid**, a framework for synthesizing proof-centric mathematical benchmarks across diverse mathematical branches, which is the approach we employed to construct AlgGeoTest. The central idea of Proof2Hybrid is to convert mathematical definitions or proposition-proof pairs into a carefully designed hybrid question format that integrates the strengths of both multiple-choice and true-or-false questions.

### 3.1 COLLECTION OF SEED ITEMS

Two types of seed items are well-suited for Proof2Hybrid to be adapted into our hybrid question format: mathematical definitions, and mathematical proposition-proof pairs. Such forms of items are abundantly present in the natural corpus of mathematics, especially in the corpus of cutting-edge mathematics, thereby furnishing Proof2Hybrid with substantial application scenarios.

### 3.2 FILTRATION OF SEED ITEMS

During the iterative refinement of Proof2Hybrid, we observed that certain seed items were poorly written, and such seed items tend to yield distractors of equally low quality. Therefore, we implemented a filtering mechanism to identify and exclude these poorly formulated seed items from the full set.

The filtering mechanism employs $m_1$ leading LLMs to assess the mathematical consistency of each seed item, with each model rendering its judgment $n_1$ times, thereby yielding a total of $m_1 n_1$ model-based verdicts per item. We retain all seed items that were adjudicated mathematically correct on at least $k_1$ occasions and exclude all others. Here $m_1, n_1, k_1$ are hyper parameters, and we enforce $k_1 > \frac{m_1 n_1}{2}$ so that retained items are deemed correct more often than incorrect.

We need not be concerned that this filtering process will inadvertently discard genuinely difficult seed items. Empirically, when the model encounters such questions——for instance, when it cannot fully comprehend a proof——it is unable to identify any flaws and consequently deems the definition or proof mathematically correct. Thus, these difficult questions will not be removed by the filter.

In our AlgGeoTest production scenario, we pick $m_1 = 4$, $n_1 = 3$, and $k_1 = 8$, and the 4 leading LLMs we employed are o3 (OpenAI, 2025a), Gemini-2.5-Pro (Google, 2025), DeepSeek-R1 (Guo et al., 2025) and Qwen3-235B-A22B (Qwen Team, 2025).

### 3.3 GENERATION OF DISTRACTORS

We employ $m_2$ LLMs for the generation of the distractors. Specifically, given each seed item, we instruct every model to craft $n_2$ close but mathematically flawed distractors by strategically altering keywords, conditions, and formulas within the statement (for proposition-proof pairs, only the proof is modified, and the proposition remains intact). We then randomly select $k_2$ distractors from the $n_2$ available, yielding $m_2 k_2$ distractors in total from the $m_2$ models. Subsequently, we perform a deduplication process to eliminate redundant distractors that become identical upon the removal of spaces and line breaks, retaining only one instance of each class. Here $m_2, n_2, k_2$ are hyper parameters, and we require $k_2 \leq n_2$.

We employ multiple models rather than a single one in order to eliminate the potential bias inherent in the adaptation performed by any individual model (Abels & Lenaerts, 2025). This approach—instructing each model first generate $n_2$ distractors before selecting $k_2$—helps minimize the probability that different models will produce mathematically identical or highly similar distractors.

In our scenario of producing AlgGeoTest, we pick $m_2 = 5$, $n_2 = 6$, and $k_2 = 2$, and the 5 LLMs we employed are DeepSeek-V3 (DeepSeek-AI et al., 2024), Qwen2.5-72B-Insturct (Qwen Team, 2024), GPT-4.1 (OpenAI, 2025c), Claude-4-Sonnet (Anthropic, 2025a), Gemini-2.5-Flash (DeepMind, 2025).

### 3.4 FILTRATION OF DISTRACTORS

Distractors generated by LLMs are often unsatisfactory: some are too easily exposed because the adaptation creates glaring inconsistencies, while others are flawed in subtler ways — leaning on external context (e.g., unreferenced theorems), cloaked in ambiguity, or even being in fact mathematically correct, thus rendering their mathematical truth or falsity logically undetermined by the model.

We employ an LLM-based filter in order to remove these unsatisfactory distractors. The configuration mirrors that used for filtering seed items: the same $m_3$ leading LLMs, each judge the mathematical correctness of every distractor $n_3$ times, producing $m_3 n_3$ independent verdicts. In contrast to the seed item stage, we now retain only those distractors deemed incorrect in $k_3$ to $k_4$ occasions and discard the rest. Here $m_3, n_3, k_3, k_4$ are hyper parameters satisfying

$$\frac{m_3 n_3}{2} < k_3 \leq k_4 \leq m_3 n_3 - 2.$$

The lower bound $\frac{m_3 n_3}{2} < k_3$ ensures that the retained distractors are deemed incorrect more often than correct, while the upper bound $k_4 \leq m_3 n_3 - 2$ guarantees that these distractors are sufficiently confusing for the models to make at least two false verdicts.

This filtering process cleanly eliminates the two types of flawed distractors described earlier. Distractors that are too easy to spot will be marked incorrect more than $k_4$ times and then be rejected, while the distractors that are logically undecidable by LLMs will be marked incorrect less than $k_3$ times, because when a model detects no mathematical inconsistency, it prefers to mark the statement as correct rather than guess.

We leverage multiple LLMs rather than relying on a single one in order to mitigate potential bias during evaluation (Abels & Lenaerts, 2025).

In our scenario of producing AlgGeoTest, we pick $m_3 = 4$, $n_3 = 3$, $k_3 = 7$, and $k_4 = 10$, and the 4 leading LLMs we employed are o3 (OpenAI, 2025a), Gemini-2.5-Pro (Google, 2025), DeepSeek-R1 (Guo et al., 2025) and Qwen3-235B-A22B (Qwen Team, 2025).

### 3.5 AGGREGATION OF HYBRID-FORMATTED QUESTIONS

#### 3.5.1 GENERATION-BASED EVALUATION

We consolidate the seed items and the model-generated distractors into our carefully designed hybrid-formatted questions. From the full pool of all seed items and distractors we repeatedly draw, at random, $m$ seed items and $n - m$ distractors, ensuring that all $n$ items stem from distinct seed items. These $n$ entries are assembled into a single hybrid-formatted question, and the process continues until the residual pool cannot form another complete question. Here $m$ and $n$ are hyper parameters and we require $0 < m < n$.

Evaluating these hybrid-formatted questions is straightforward: we present the model with all $n$ items and ask it to judge the mathematical correctness of each item, under the constraint that exactly $m$ of these $n$ items are correct.

We require all $n$ items of a hybrid-formatted question to originate from distinct seed items to prevent LLMs from inferring the correct answer by comparing items that share the same origin. This question format offers an additional benefit: since the task reduces to ranking the relative mathematical correctness of all items instead of classifying the absolute correctness of each item, model-specific biases arising from different mathematical correctness judgment standards of different models are mitigated.

In our scenario of producing AlgGeoTest, we pick $m = 2$ and $n = 6$. This choice is motivated by three considerations. First, having two instead of one correct answers raises the question's difficulty, as a model needs to figure out both correct answers to get scores. Second, after filtering, the ratio of seed items to model-generated distractors is approximately 1:2; this configuration ensures almost every seed item and every distractor is used, eliminating possible waste. Third, we want to keep the context length of each question within a reasonable range, hence the value of $n$ should not be too large. However, alternative choices of $m$ and $n$ are also welcomed to tailor generated questions to the desired difficulty.

We include the prompts we used for generation-based evaluation in Appendix A.

#### 3.5.2 PERPLEXITY-BASED EVALUATION

We propose a perplexity-based evaluation protocol, rather than the conventional generation-based one, to reduce the cognitive burden on LLMs—an advantage especially pronounced when assessing base models.

To apply this evaluation protocol, we abandon the hybrid question format and switch to a standard multiple-choice design. Each seed item and its derived distractors are packaged into one question. During evaluation, we compute the perplexity of every option—including the seed item and all distractors—and select the option with the lowest perplexity as the model's final answer.

### 3.6 ANALYSIS OF HYBRID QUESTION FORMAT

In this section we provide a deep analysis of why our hybrid question format eliminates the disadvantages of choice questions and true-or-false questions, that is, our hybrid question format are hard to guess, reject comparison between different options, and relieve the bias of mathematical correctness standards between different LLMs. A pros and cons comparison between true-or-false, multiple choice, and our hybrid question format is presented in Figure 3. An example demonstrating the drawbacks of multiple-choice question format is given in Appendix C.

The first two advantages of our hybrid question format are straightforward: on the one hand, random guessing gain an expected accuracy of $1/C_n^m$, which becomes lower than even $0.1$ with small choices of $m$ and $n$ such as $m = 2, n = 6$; one the other hand, since we require all options of a question to stem from different items, it is impossible for LLMs to guess the correct answer by simply comparing between different options.

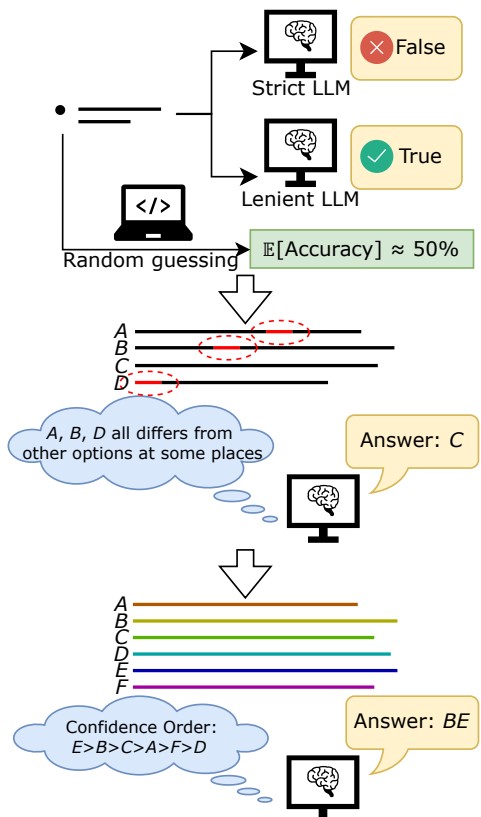

Figure 3: Pros and Cons comparison between true-or-false questions, multiple choice questions, and our hybrid question format. An example demonstrating the drawbacks of multiple-choice question format is given in Appendix C.

The third advantage is subtler. At the beginning we have various mathematical definitions and proposition-proof pairs, we then input it into LLMs and ask them to adjust some part of the mathematical statements to generate similar but incorrect counterparts of the statements. After going through a filtering process, these generated statements are mixed up with original ones and randomly distributed into groups of $n$, each group with $m$ correct terms from the original items and $n - m$ incorrect terms from the generated items. Vary as the standards of correctness may across models, a term that belongs to the original correct items should always be "more correct" than a term from the generated twisted items. Since each group of $n$ guarantees to have $m$ terms that are "more correct" than the rest $n - m$ ones, when we ask a model to choose $m$ "most correct" terms from $n$, the bias brought by this different standard is eliminated.

## 4 EXPERIMENTS

### 4.1 GENERATION OF BENCHMARK

We instantiate Proof2Hybrid on the definitions and propositions of the open source Algebraic Geometry textbook and reference work "The Stacks project" (Stacks-Project-Authors, 2025)—a comprehensive reference for algebraic geometry and related topics spanning from undergraduate foundations to the research frontier—producing **AlgGeoTest**. We decided to use "The Stacks project" since its data format is perfectly suitable for Proof2Hybrid: in "The Stacks project", each definition, theorem, proposition or lemma is contained in a tag, and from these tags, we can easily extract mathematical definitions or math proposition-proof pairs, which can be directly used as seed items for Proof2Hybrid. We randomly selected 1,100 seed items from the full set for the implementation of Proof2Hybrid.

A sample problem from AlgGeoTest is presented in Appendix B.

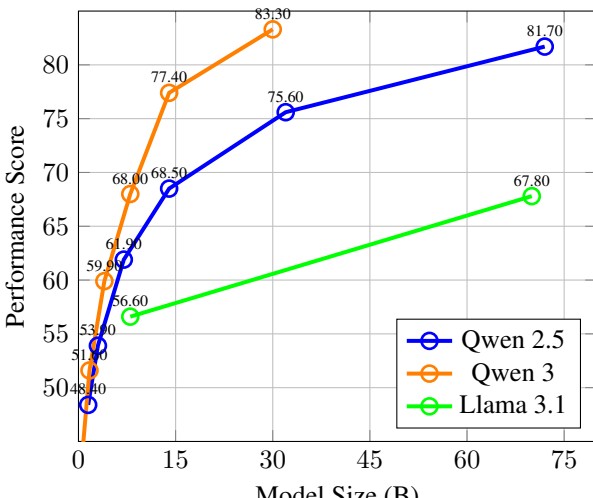

Figure 4: Performance evaluation across families of base models. The plots demonstrate the scaling behavior of (a) Qwen 2.5, (b) Qwen 3, (c) Llama3.1 families across different parameter scales. Each subplot shows the performance trend as model size increases, with individual data points annotated for clarity.

## 4.2 EVALUATION RESULTS

### 4.2.1 GENERATION-BASED EVALUATION

We evaluate AlgGeoTest on both open-sourced and closed-sourced LLMs, including Gemini-2.5-Pro, Gemini-2.5-Flash, o3, o4-mini, GPT-4.1, GPT-4o, Claude-4-Sonnet, Claude-4-Opus, Grok-4, DeepSeek-R1, DeepSeek-V3, DeepSeek-R1-Distill-Qwen-32B, Qwen2.5-72B-Instruct, Qwen3-235B-A22B, Qwen3-30B-A3B, Qwen3-32B, QwQ-32B, Kimi-K2 (Google, 2025; DeepMind, 2025; OpenAI, 2025a;b;c; 2024; Anthropic, 2025a;b; xAI , Elon Musk's AI Company; Guo et al., 2025; DeepSeek-AI et al., 2024; DeepSeek-AI, 2025; Qwen Team, 2024; 2025; Alibaba Cloud Qwen Team, 2025b; Yang et al., 2025; Alibaba Cloud Qwen Team, 2025a; AI, 2025).

All evaluations are based on API, and use the default hyper parameters of each service.

We adopt two grading metrics: a loose metric and a tight one. The loose metric awards full credit when both model answers are correct, half credit when exactly one is correct, and zero otherwise. The tight metric grants full credit only if both answers are entirely correct and assigns zero in all other cases.

The evaluation results are shown in Figure 2. The evaluation results reveal that even the best-performing model to date achieves only a moderate score of around 60, with scores commonly lower than 20. This result underscores the rigorous and challenging nature of our benchmark. We also observe that reasoning models commonly outperform their non-reasoning counterparts, demonstrating that our benchmark effectively measures the reasoning capability of LLMs.

### 4.2.2 PERPLEXITY-BASED EVALUATION

We evaluate AlgGeoTest on open-sourced base models including the Qwen2.5 family, the Qwen3 family and the Llama3.1 family (Yang et al., 2024; 2025; AI, 2024), using the perplexity-based evaluation protocol. Since each multiple-choice question has a different number of options and therefore a different level of difficulty, we assigned a weighted score to every question so that, while keeping the total maximum score at 100, the expected points gained by random guessing are the same for each question. The evaluation results are shown in Figure 4. As we can see, the scores of

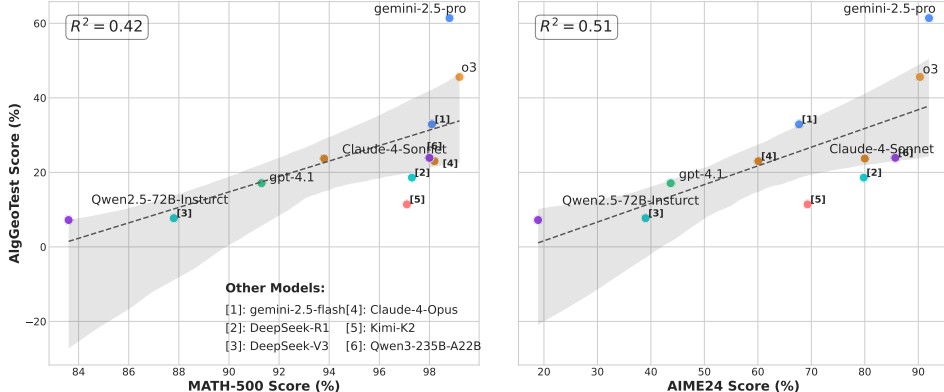

Figure 5: Two scatter plots comparing the performance of various LLMs on AlgGeoTest versus MATH-500 and AIME24. Both plots include a linear regression line with a shaded confidence interval, and the R-squared value is displayed in the top-left corner of each plot. As we can see, the R-squared value of AlgGeoTest versus MATH-500 is 0.42, and the R-squared value of AlgGeoTest versus AIME24 is 0.51. This result indicates that, while all three benchmarks reside within the broader domain of mathematics, AlgGeoTest examines a subfield distinct from those targeted by MATH-500 and AIME24.

these models increase with model size, demonstrating strong scaling properties, providing evidence of the robustness of our benchmark.

### 4.3 COMPARISON WITH MAINSTREAM BENCHMARKS

We compare the evaluation results on AlgGeoTest with those on mainstream mathematical benchmarks such as MATH-500 (Hendrycks et al., 2021) and AIME24 (, Maxwell-Jia). The results are demonstrated in Figure 5. As we can see, the scores on AlgGeoTest exhibit a correlation—though not a strictly linear one—with those on MATH-500 and AIME24. This indicates that, while all three benchmarks reside within the broader domain of mathematics, AlgGeoTest examines a subfield distinct from those targeted by MATH-500 and AIME24. Comparison of AlgGeoTest with more mainstream mathematical benchmark is provided in Appendix D.

### 4.4 AUDIT OUTCOMES

To safeguard quality, we engaged expert mathematicians to audit every question of AlgGeoTest and the responses produced by leading models. The audit reveals that over 98.75% of model-generated distractors are mathematically incorrect yet deceptively plausible, while more than 95% of the benchmark questions meet the same standard, with every distractor satisfying the same stringent criteria.

Model failures on AlgGeoTest arise chiefly from two sources: an inability to detect subtle flaws in distractors or a hallucinated belief that a valid item is inconsistent. Both shortcomings trace to intrinsic limitations and gaps in background knowledge, not to any defect in AlgGeoTest itself. The benchmark's uncompromising quality thus offers compelling evidence for the robustness of Proof2Hybrid.

### 5 CONCLUSION

In this paper, we propose Proof2Hybrid, the first fully automated framework for synthesizing proof-centric mathematical benchmarks. Using Proof2Hybrid, we construct AlgGeoTest, a benchmark consisting of 456 items designed to evaluate large language models' understanding of algebraic geometry. Experimental results along with follow-up manual review confirm the high quality of our benchmark and thus demonstrate the robustness of Proof2Hybrid, utilizing which we are able to produce scalable generation of similar benchmarks across a broad spectrum of mathematical domains.

## 6 REPRODUCIBILITY STATEMENT

Our code and data for the pipeline of Proof2Hybrid are provided in the supplementary material in order to facilitate reproducibility.

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

## A PROMPT USED FOR GENERATION-BASED EVALUATION

We present below the prompt we used for the generation based evaluation of AlgGeoTest:

**Prompt Used for Generation-Based Evaluation**

```
def generate_one_choice(s):
    if s["type"] == "definition":
        prompt = "This choice is a mathematical definition. Here is
            the definition:\n" + s["text"]
    else:
        prompt = "This choice is a mathematical proposition-proof
            pair. Here is the proposition:\n" + s["proposition"] +"\
            n\nHere is the proof of the proposition:\n" + s["text"]
    return prompt

def generate_one_request(data):
    prompt = """
    Below is a choice question, each choice is either a mathematical
        definition or a mathematical proposition-proof pair. Your
        goal is to judge the mathematical correctness of each choice
        (for proposition-proof pairs, this means the correctness of
        the proof, and the proposition is always assumed to be
        correct), and find the correct choices. Only two choices
        among all choices are mathematically correct. Please think
        step by step and find the two mathematically correct choices
        .

    When judging the correctness of the choices, you should only
        focus on whether the mathematics and logic in it are correct
        , and your judge should not be influenced by those non-
        mathematical things. In particular, your judge should not be
         influenced by things related to references such as things
        inside a \\ref{}, or the index of a refered lemma.

    When judging the correctness of the choices, you should be
        primarily focused on whether there exist mathematical or
        logical inconsistency, and mathematical completeness is of
        secondary importance. This means even a typo should be
        considered incorrect if it make the definition or proof
        inconsistent, and some minor omission of the proof that do
        not affect the consistency should not be considered
        incorrect.

    Output format: you should put the labels of the two choices that
         you think are correct inside a \\boxed{}, and put it at the
         end of your output. For example, you should return \\boxed{
        A,E} if you think the two correct choices are A and E, and
        you should return \\boxed{C,F} if you think the two correct
        choices are C and F.

    Here are the choices:
    """ + "\n\n\nChoice A:\n\n" + generate_one_choice(data["A"]) + "
        \n\n\nChoice B:\n\n" + generate_one_choice(data["B"]) + "\n\
        n\nChoice C:\n\n" + generate_one_choice(data["C"]) + "\n\n\
        nChoice D:\n\n" + generate_one_choice(data["D"]) +"\n\n\
        nChoice E:\n\n" + generate_one_choice(data["E"]) + "\n\n\
        nChoice F:\n\n" + generate_one_choice(data["F"])
    return [
            {"role": "user", "content": prompt}
        ]
```

## B  A SAMPLE PROBLEM FROM ALGGEOTEST

We include below an example problem from AlgGeoTest, the responses of it from Gemini-2.5-Pro and o3, and their error analysis.

### B.1  THE 6 OPTIONS

We first present the 6 options of the sample problem below:

---

**Option A**

**Tag of The Stacks project:** 04Z8

**Type:** Proposition-proof pair.

**Ground Truth:** False.

**Proposition:** Let $f : X \to Y$ be a continuous map of topological spaces.

1. If $X$ is Noetherian, then $f(X)$ is Noetherian.
2. If $X$ is locally Noetherian and $f$ open, then $f(X)$ is locally Noetherian.

**Original Proof:** In case (1), suppose that $Z_1 \supset Z_2 \supset Z_3 \supset \ldots$ is a descending chain of closed subsets of $f(X)$ (as usual with the induced topology as a subset of $Y$). Then $f^{-1}(Z_1) \supset f^{-1}(Z_2) \supset f^{-1}(Z_3) \supset \ldots$ is a descending chain of closed subsets of $X$. Hence this chain stabilizes. Since $f(f^{-1}(Z_i)) = Z_i$ we conclude that $Z_1 \supset Z_2 \supset Z_3 \supset \ldots$ stabilizes also. In case (2), let $y \in f(X)$. Choose $x \in X$ with $f(x) = y$. By assumption there exists a neighbourhood $E \subset X$ of $x$ which is Noetherian. Then $f(E) \subset f(X)$ is a neighbourhood which is Noetherian by part (1).

**Adapted Proof:** In case (1), suppose that $Z_1 \supset Z_2 \supset Z_3 \supset \ldots$ is a descending chain of closed subsets of $f(X)$ (as usual with the induced topology as a subset of $Y$). Then $f^{-1}(Z_1) \supset f^{-1}(Z_2) \supset f^{-1}(Z_3) \supset \ldots$ is a descending chain of closed subsets of $X$. Hence this chain stabilizes. Since $f(f^{-1}(Z_i)) = Z_i$ we conclude that $Z_1 \supset Z_2 \supset Z_3 \supset \ldots$ stabilizes also. In case (2), let $y \in f(X)$. Choose $x \in X$ with $f(x) = y$. By assumption there exists a neighbourhood $E \subset X$ of $x$ which is locally Noetherian. Then $f(E) \subset f(X)$ is a neighbourhood which is Noetherian by part (1).

**Difference Between Original Proof and Adapted Proof:** The adapted proof changes Noetherian to locally Noetherian. This is a mathematically incorrect change.

---

## Option B

**Tag of The Stacks project:** 0B3M

**Type:** Proposition-proof pair.

**Ground Truth:** False.

**Proposition:** Let $X$ be a scheme. Let $\mathcal{F}$ be a quasi-coherent $\mathcal{O}_X$-module. If $U \subset X$ is open and WeakAss$(\mathcal{F}) \subset U$, then $\Gamma(X, \mathcal{F}) \to \Gamma(U, \mathcal{F})$ is injective.

**Original Proof:** Let $s \in \Gamma(X, \mathcal{F})$ be a section which restricts to zero on $U$. Let $\mathcal{F}' \subset \mathcal{F}$ be the image of the map $\mathcal{O}_X \to \mathcal{F}$ defined by $s$. Then Supp$(\mathcal{F}') \cap U = \emptyset$. On the other hand, WeakAss$(\mathcal{F}') \subset$ WeakAss$(\mathcal{F})$ by Lemma \ref{divisors-lemma-ses-weakly-ass}. Since also WeakAss$(\mathcal{F}') \subset$ Supp$(\mathcal{F}')$ (Lemma \ref{divisors-lemma-weakly-ass-support}) we conclude WeakAss$(\mathcal{F}') = \emptyset$. Hence $\mathcal{F}' = 0$ by Lemma \ref{divisors-lemma-weakly-ass-zero}.

**Adapted Proof:** Let $s \in \Gamma(X, \mathcal{F})$ be a section which restricts to zero on $U$. Let $\mathcal{F}' \subset \mathcal{F}$ be the image of the map $\mathcal{O}_X \to \mathcal{F}$ defined by $s$. Then Supp$(\mathcal{F}') \cap U = \emptyset$. On the other hand, Ass$(\mathcal{F}') \subset$ Ass$(\mathcal{F})$ by Lemma \ref{divisors-lemma-ses-weakly-ass}. Since also Ass$(\mathcal{F}') \subset$ Supp$(\mathcal{F}')$ (Lemma \ref{divisors-lemma-weakly-ass-support}) we conclude Ass$(\mathcal{F}') = \emptyset$. Hence $\mathcal{F}' = 0$ by Lemma \ref{divisors-lemma-weakly-ass-zero}.

**Difference Between Original Proof and Adapted Proof:** The adapted proof changes WeakAss to Ass. This is a mathematically incorrect change.

## Option C

**Tag of The Stacks project:** 08LR

**Type:** Proposition-proof pair.

**Ground Truth:** True.

**Proposition:** Let $\mathcal{C}$ be a category, and let $f : X \to Y$ be a morphism of $\mathcal{C}$. Then

1. $f$ is a monomorphism if and only if $X$ is the fibre product $X \times_Y X$, and
2. $f$ is an epimorphism if and only if $Y$ is the pushout $Y \amalg_X Y$.

**Original Proof:** Let suppose that $f$ is a monomorphism. Let $W$ be an object of $\mathcal{C}$ and $\alpha, \beta \in \mathrm{Mor}_{\mathcal{C}}(W, X)$ such that $f \circ \alpha = f \circ \beta$. Therefore $\alpha = \beta$ as $f$ is monic. In addition, we have the commutative diagram

$$
\begin{array}{ccc}
X & \xrightarrow{\mathrm{id}_X} & X \\
\mathrm{id}_X \downarrow & & \downarrow f \\
X & \xrightarrow{f} & Y
\end{array}
$$

which verify the universal property with $\gamma := \alpha = \beta$. Thus $X$ is indeed the fibre product $X \times_Y X$.

Suppose that $X \times_Y X \cong X$. The diagram

$$
\begin{array}{ccc}
X & \xrightarrow{\mathrm{id}_X} & X \\
\mathrm{id}_X \downarrow & & \downarrow f \\
X & \xrightarrow{f} & Y
\end{array}
$$

commutes and if $W \in \mathrm{Ob}(\mathcal{C})$ and $\alpha, \beta : W \to X$ such that $f \circ \alpha = f \circ \beta$, we have a unique $\gamma$ verifying

$$
\gamma = \mathrm{id}_X \circ \gamma = \alpha = \beta
$$

which proves that $\alpha = \beta$.

The proof is exactly the same for the second point, but with the pushout $Y \amalg_X Y = Y$.

**Option D**

**Tag of The Stacks project:** 0C0L

**Type:** Proposition-proof pair.

**Ground Truth:** False.

**Proposition:** Let $f : X \to S$ be a proper morphism with geometrically connected fibres. Let $s'' \rightsquigarrow s' \rightsquigarrow s$ be specializations of points of $S$. A composition of specialization maps $\pi_1(X_{\overline{s}''}) \to \pi_1(X_{\overline{s}'}) \to \pi_1(X_{\overline{s}})$ is a specialization map $\pi_1(X_{\overline{s}''}) \to \pi_1(X_{\overline{s}})$.

**Original Proof:** Let $\mathcal{O}_{S,s} \to A$ be the strict henselization constructed using $\kappa(s) \to \kappa(\overline{s})$. Let $A \to \kappa(\overline{s}')$ be the map used to construct the first specialization map. Let $\mathcal{O}_{S,s'} \to A'$ be the strict henselization constructed using $\kappa(s') \subset \kappa(\overline{s}')$. By functoriality of strict henselization, there is a map $A \to A'$ such that the composition with $A' \to \kappa(\overline{s}')$ is the given map (Algebra, Lemma \ref{algebra-lemma-map-into-henselian-colimit}). Next, let $A' \to \kappa(\overline{s}'')$ be the map used to construct the second specialization map. Then it is clear that the composition of the first and second specialization maps is the specialization map $\pi_1(X_{\overline{s}''}) \to \pi_1(X_{\overline{s}})$ constructed using $A \to A' \to \kappa(\overline{s}'')$.

**Adapted Proof:** Let $\mathcal{O}_{S,s} \to A$ be the strict henselization constructed using $\kappa(s) \to \kappa(\overline{s})$. Let $A \to \kappa(\overline{s}')$ be the map used to construct the first specialization map. Let $\mathcal{O}_{S,s'} \to A'$ be the strict henselization constructed using $\kappa(s') \subset \kappa(\overline{s}')$. Using functoriality of strict henselization, we obtain a map $A' \to A$ such that its composition with $A \to \kappa(\overline{s}')$ is the given map. Next, let $A' \to \kappa(\overline{s}'')$ be the map used to construct the second specialization map. Then we conclude that the composition of the first and second specialization maps is the specialization map $\pi_1(X_{\overline{s}''}) \to \pi_1(X_{\overline{s}})$ constructed using $A' \to A \to \kappa(\overline{s}'')$.

**Difference Between Original Proof and Adapted Proof:** The adapted proof switches the direction of the arrow $A \to A'$, changing it to $A' \to A$. This is a mathematically incorrect change.

**Option E**

**Tag of The Stacks project:** 0EUD

**Type:** Proposition-proof pair.

**Ground Truth:** True.

**Proposition:** Let $\{f_i : X_i \to X\}_{i \in I}$ be a family of morphisms of affine schemes. Assume the equivalent assumption of Lemma \ref{descent-lemma-universal-effective-epimorphism-affine} hold and that moreover for any morphism of affines $Y \to X$ the map

$$\coprod X_i \times_X Y \longrightarrow Y$$

is a submersive map of topological spaces (Topology, Definition \ref{topology-definition-submersive}). Then our family of morphisms is a universal effective epimorphism in the category of schemes.

**Original Proof:** By Lemma \ref{descent-lemma-check-universal-effective-epimorphism-affine} it suffices to base change our family of morphisms by $Y \to X$ with $Y$ affine. Set $Y_i = X_i \times_X Y$. Let $T$ be a scheme and let $h_i : Y_i \to T$ be a family of morphisms such that $h_i \circ \mathrm{pr}_1 = h_j \circ \mathrm{pr}_2$ on $Y_i \times_Y Y_j$. Note that $Y$ as a set is the coequalizer of the two maps from $\coprod Y_i \times_Y Y_j$ to $\coprod Y_i$. Namely, surjectivity by the affine case of Lemma \ref{descent-lemma-universal-effective-epimorphism-surjective} and injectivity by Lemma \ref{descent-lemma-equiv-fibre-product}. Hence there is a set map of underlying sets $h : Y \to T$ compatible with the maps $h_i$. By the second condition of the lemma we see that $h$ is continuous! Thus if $y \in Y$ and $U \subset T$ is an affine open neighbourhood of $h(y)$, then we can find an affine open $V \subset Y$ such that $h(V) \subset U$. Setting $V_i = Y_i \times_Y V = X_i \times_X V$ we can use the result proved in Lemma \ref{descent-lemma-universal-effective-epimorphism-affine} to see that $h|_V : V \to U \subset T$ comes from a unique morphism of affine schemes $h_V : V \to U$ agreeing with $h_i|_{V_i}$ as morphisms of schemes for all $i$. Glueing these $h_V$ (see Schemes, Section \ref{schemes-section-glueing-schemes}) gives a morphism $Y \to T$ as desired.

**Option F**

**Tag of The Stacks project:** 0BI9

**Type:** Definition.

**Ground Truth:** False.

**Original Definition:** Let $X$ be a locally Noetherian scheme. A *strict normal crossings divisor* on $X$ is an effective Cartier divisor $D \subset X$ such that for every $p \in D$ the local ring $\mathcal{O}_{X,p}$ is regular and there exists a regular system of parameters $x_1, \ldots, x_d \in \mathfrak{m}_p$ and $1 \le r \le d$ such that $D$ is cut out by $x_1 \ldots x_r$ in $\mathcal{O}_{X,p}$.

**Adapted Definition:** Let $X$ be a locally Noetherian scheme. A *strict normal crossings divisor* on $X$ is an effective Cartier divisor $D \subset X$ such that for every $p \in D$ the local ring $\mathcal{O}_{X,p}$ is regular and there exists a regular system of parameters $x_1, \ldots, x_d \in \mathfrak{m}_p$ and $1 \le r < d$ such that $D$ is cut out by $x_1 \ldots x_r$ in $\mathcal{O}_{X,p}$.

**Difference Between Original Definition and Adapted Definition:** The adapted definition changes $1 \le r \le d$ to $1 \le r < d$. This is a mathematically incorrect change.

## B.2 GEMINI-2.5-PRO AND O3'S RESPONSES

We now present Gemini-2.5-Pro and o3's responses below:

### B.2.1 GEMINI-2.5-PRO'S RESPONSE

An analysis of each choice is as follows:

**Choice A:** The proposition consists of two parts. 1. If $f : X \to Y$ is a continuous map and $X$ is a Noetherian topological space, then $f(X)$ is a Noetherian topological space. 2. If $X$ is locally Noetherian and $f$ is an open map, then $f(X)$ is locally Noetherian.

The proof for part (1) is correct. A descending chain of closed subsets $Z_n$ in $f(X)$ gives a descending chain of closed subsets $f^{-1}(Z_n)$ in $X$. Since $X$ is Noetherian, the chain of preimages stabilizes. As $f$ is surjective onto its image, $f(f^{-1}(Z_n)) = Z_n$, so the chain of $Z_n$ also stabilizes.

The proof for part (2) has a flaw. It states: "By assumption there exists a neighbourhood $E \subset X$ of $x$ which is locally Noetherian." The assumption is that $X$ is a locally Noetherian space. By definition, this means for any point $x \in X$, there exists a neighborhood of $x$ which is a *Noetherian space*. A Noetherian space is locally Noetherian, but the converse is not true. The proof should have stated that we can choose a *Noetherian* neighborhood $E$ of $x$. Then it continues: "Then $f(E) \subset f(X)$ is a neighbourhood which is Noetherian by part (1)". To apply part (1), the domain of the map must be Noetherian. The proof claims a "locally Noetherian" neighborhood is sufficient, which is incorrect. The image of a locally Noetherian space under a continuous map is not in general Noetherian. This is a logical error in the proof. Thus, choice A is incorrect.

**Choice B:** The proposition is a known result in scheme theory. Let's examine the proof. The proof considers a section $s \in \Gamma(X, \mathcal{F})$ restricting to zero on $U$ and defines a subsheaf $\mathcal{F}'$ as the image of the map $\mathcal{O}_X \to \mathcal{F}$ induced by $s$. The steps are: 1. $\mathrm{Supp}(\mathcal{F}') \cap U = \emptyset$. This is correct, as $s|_U = 0$ means $s_x = 0$ for all $x \in U$. 2. It claims $\mathrm{Ass}(\mathcal{F}') \subset \mathrm{Ass}(\mathcal{F})$ and $\mathrm{Ass}(\mathcal{F}') \subset \mathrm{Supp}(\mathcal{F}')$. These inclusions are correct. 3. It concludes $\mathrm{Ass}(\mathcal{F}') = \emptyset$. This follows from $\mathrm{WeakAss}(\mathcal{F}) \subset U$ and the observations above (assuming Ass means WeakAss). 4. The final step is "Hence $\mathcal{F}' = 0$ by Lemma...". This relies on the property that if $\mathrm{Ass}(\mathcal{F}') = \emptyset$, then $\mathcal{F}' = 0$. This property is true for quasi-coherent sheaves on *locally Noetherian* schemes. However, the proposition is stated for an arbitrary scheme $X$. For a general (non-Noetherian) ring $A$, a non-zero $A$-module can have an empty set of associated primes. Thus, a non-zero quasi-coherent sheaf on a non-locally-Noetherian scheme may have an empty set of associated points (in the usual sense of prime ideals that are annihilators of elements). The correct statement for general schemes uses weakly associated points: $\mathrm{WeakAss}(\mathcal{F}') = \emptyset$ implies $\mathcal{F}' = 0$. The proof explicitly uses Ass, not WeakAss. This makes the proof mathematically incorrect as it applies a result beyond its scope of validity. Therefore, choice B is incorrect.

**Choice C:** The proposition gives categorical characterizations of monomorphisms and epimorphisms. 1. A morphism $f : X \to Y$ is a monomorphism if and only if the diagonal morphism $\Delta_f : X \to X \times_Y X$ is an isomorphism. The statement "$X$ is the fibre product $X \times_Y X$" is a common, though imprecise, way of saying this, with the understanding that the projection maps from $X$ to $X$ are the identity maps. 2. Dually, $f$ is an epimorphism iff the codiagonal map $\nabla_f : Y \amalg_X Y \to Y$ is an isomorphism. Again, "$Y$ is the pushout $Y \amalg_X Y$" is the corresponding shorthand.

The proof spells out this equivalence. - For (1): $f$ is a monomorphism iff for any $W$, and morphisms $\alpha, \beta : W \to X$, $f \circ \alpha = f \circ \beta \implies \alpha = \beta$. The universal property of the fiber product $P = X \times_Y X$ with projections $p_1, p_2$ says that for any such pair $\alpha, \beta$, there exists a unique $\gamma : W \to P$ with $p_1 \gamma = \alpha, p_2 \gamma = \beta$. If $P = X$ and $p_1 = p_2 = \mathrm{id}_X$, this property becomes: there is a unique $\gamma : W \to X$ such that $\gamma = \alpha$ and $\gamma = \beta$. This forces $\alpha = \beta$. Conversely, if $f$ is a monomorphism, this condition holds, so we can define the unique map $\gamma = \alpha = \beta$, so the universal property is satisfied. The logic is sound, even if the phrasing in the proof is somewhat confusing (e.g., "$\gamma := \alpha = \beta$"). - For (2): The statement is dual to (1), and the proof is dually the same. The mathematical content is correct. Thus, choice C is correct.

**Choice D:** The proposition is about the composition of specialization maps for fundamental groups. The proof describes the construction of these maps using strict henselizations of local rings.

A specialization of points $s' \rightsquigarrow s$ in $S$ corresponds to a local ring map $\mathcal{O}_{S,s} \to \mathcal{O}_{S,s'}$. Let $A = \mathcal{O}_{S,s}^{sh}$ and $A' = \mathcal{O}_{S,s'}^{sh}$ be the strict henselizations at $s$ and $s'$ with respect to choices of geometric points. The functoriality of strict henselization provides a map $A \to A'$ (from the strict henselization at the more general point to that at the more special point). The proof states: "Using functoriality of strict henselization, we obtain a map $A' \to A$". This is the reverse of the correct direction of the map. This fundamental error invalidates the rest of the argument. Thus, choice D is incorrect.

**Choice E:** This proposition provides sufficient conditions for a family of morphisms of affine schemes to be a universal effective epimorphism in the category of all schemes. The proof is a standard argument in descent theory. Let $\{f_i : X_i \to X\}$ be the family. The proof first correctly reduces the problem, via a standard lemma, to showing that for any affine scheme $Y \to X$, the base-changed family $\{Y_i \to Y\}$ is an effective epimorphism in the category of schemes. Given a descent datum $(h_i : Y_i \to T)$, the proof correctly pieces together a morphism $h : Y \to T$. 1. It uses the property of being an effective epimorphism in affine schemes to deduce that there's a unique map of sets $|h| : |Y| \to |T|$. 2. It correctly uses the "submersive" hypothesis to show that this map of sets is continuous. The quotient topology on $|Y|$ induced by $|\coprod Y_i| \to |Y|$ guarantees that $|h|$ is continuous because $|h| \circ |\coprod p_i| = |\coprod h_i|$ is continuous. 3. It then shows that the continuous map $h$ is a morphism of schemes by checking it locally. For any affine open cover of $T$, the preimages in $Y$ can be covered by affine opens $V \subset Y$. Restricting the descent datum to $V$ (i.e. to $\{V_i \to V\}$), one can use the hypothesis (that the family is an effective epimorphism for affines) to get a unique morphism $V \to U$. 4. These locally defined morphisms glue together by uniqueness to give the desired global morphism $h : Y \to T$. The argument is detailed and logically sound. Thus, choice E is correct.

**Choice F:** This choice provides a definition for a "strict normal crossings divisor". The key parts of the definition are that at any point $p \in D$, the local ring $\mathcal{O}_{X,p}$ is regular, and the local equation of $D$ is $x_1 \ldots x_r$ for some elements $x_1, \ldots, x_r$ that are part of a regular system of parameters $x_1, \ldots, x_d$. The controversial part is the condition $1 \le r < d$. The standard definition of a (simple) normal crossings divisor requires $1 \le r \le d$. The condition $r < d$ excludes many standard examples. For instance: - In a 2-dimensional space, the divisor given by the intersection of two coordinate axes (e.g. $xy = 0$ in $\mathbb{A}^2$) would have $r = 2, d = 2$. This is excluded by $r < d$. - On a curve, the local ring at any point has dimension $d = 1$. The condition $1 \le r < 1$ is impossible. This would mean that there are no strict normal crossings divisors on any curve, which contradicts the existence of even smooth divisors (where $r = 1$). This non-standard and restrictive condition makes the definition mathematically problematic and inconsistent with common usage and basic examples. Therefore, choice F is incorrect.

Based on the analysis, choices C and E are the two mathematically correct choices.

### B.2.2 O3'S RESPONSE

After analysing each choice for mathematical consistency:

• A: Part (2) applies Proposition (1) to an open neighbourhood declared only "locally Noetherian", not necessarily Noetherian; hence the proof is invalid.

• B: The argument shows $\mathrm{Ass}(\mathcal{F}') = \emptyset$ but then invokes a lemma that needs $\mathrm{WeakAss}(\mathcal{F}') = \emptyset$. $\mathrm{Ass}(\mathcal{F}') = \emptyset$ does not imply $\mathrm{WeakAss}(\mathcal{F}') = \emptyset$, so the proof is flawed.

• C: The characterisations of monomorphisms via pull-backs and of epimorphisms via push-outs are proved correctly. The arguments verify the universal properties in both directions with no logical gaps.

• D: The claimed map of strict henselisations should go $A \to A'$ (induced by $\mathcal{O}_{S,s} \to \mathcal{O}_{S,s'}$), but the proof needs $A' \to A$. Hence the direction of the arrow is wrong and the proof fails.

• E: Several delicate topological and sheaf-theoretic steps depend on earlier lemmas; however, the proof misses a uniqueness argument when gluing the local morphisms, so the universal property is not fully established.

• F: The definition of a strict normal crossings divisor is self-consistent and contains no logical contradiction.

Thus, the two mathematically correct choices are C and F.

$$\boxed{C, F}$$

### B.3 ERROR ANALYSIS

We give a brief error analysis of the responses of Gemini-2.5-Pro and o3 below:

Gemini-2.5-Pro successfully recognizes all inconsistencies of the incorrect options, and consequently gives a completely correct final answer. However, o3 does not recognize the inconsistency of option F, and consequently only gives a half-correct final answer.

## C AN EXAMPLE OF MULTIPLE-CHOICE QUESTION FORMAT

We give an example of questions synthesized if using multiple-choice question format instead of our hybrid question format in the Proof2Hybrid pipeline, demonstrating the drawbacks of multiple-choice question format and explaining why we've decided not to use this question format.

We first give the 8 options of the question, then the response from Qwen2.5-Instruct-72B, and finally a brief analysis of its response.

### C.1 THE 8 OPTIONS

We present the proposition and the 8 options of the example problem below (among them option C is the correct option):

---

**The Proposition**

If the Dedekind different of $A \to B$ is defined, then there is a canonical isomorphism $\mathcal{L}_{B/A} \to \omega_{B/A}$.

---

**Option A**

Recall that $\omega_{B/A} = \mathrm{Hom}_A(B, A)$ as $A \to B$ is finite. We send $x \in \mathcal{L}_{B/A}$ to the map $b \mapsto \mathrm{Trace}_{L/K}(bx)$. Conversely, given a $B$-linear map $\varphi : B \to A$ we obtain a $K$-linear map $\varphi_K : L \to K$. Since $K \to L$ is finite étale, we see that the trace pairing is nondegenerate (Lemma \refdiscriminant-lemma-discriminant) and hence there exists a $x \in L$ such that $\varphi_K(y) = \mathrm{Trace}_{L/K}(xy)$ for all $y \in L$. Then $x \in \mathcal{L}_{B/A}$ maps to $\varphi$ in $\omega_{B/A}$.

---

**Option B**

Recall that $\omega_{B/A} = \mathrm{Hom}_A(B, A)$ as $A \to B$ is finite. We send $x \in \mathcal{L}_{B/A}$ to the map $b \mapsto \mathrm{Trace}_{L/K}(bx)$. Conversely, given an $A$-linear map $\varphi : B \to A$ we obtain a $K$-linear map $\varphi_K : L \to K$. Since $K \to L$ is finite étale, we see that the trace pairing is nondegenerate (Lemma \refdiscriminant-lemma-discriminant) and hence there exists a $x \in L$ such that $\varphi_K(y) = \mathrm{Trace}_{L/K}(xy)$ for all $y \in B$. Then $x \in \mathcal{L}_{B/A}$ maps to $\varphi$ in $\omega_{B/A}$.

---

**Option C**

Recall that $\omega_{B/A} = \mathrm{Hom}_A(B, A)$ as $A \to B$ is finite. We send $x \in \mathcal{L}_{B/A}$ to the map $b \mapsto \mathrm{Trace}_{L/K}(bx)$. Conversely, given an $A$-linear map $\varphi : B \to A$ we obtain a $K$-linear map $\varphi_K : L \to K$. Since $K \to L$ is finite étale, we see that the trace pairing is nondegenerate (Lemma \refdiscriminant-lemma-discriminant) and hence there exists a $x \in L$ such that $\varphi_K(y) = \mathrm{Trace}_{L/K}(xy)$ for all $y \in L$. Then $x \in \mathcal{L}_{B/A}$ maps to $\varphi$ in $\omega_{B/A}$.

**Option D**

Recall that $\omega_{B/A} = \mathrm{Hom}_A(B, B)$ as $A \to B$ is finite. We send $x \in \mathcal{L}_{B/A}$ to the map $b \mapsto \mathrm{Trace}_{L/K}(bx)$. Conversely, given an $A$-linear map $\varphi : B \to A$ we obtain a $K$-linear map $\varphi_K : L \to K$. Since $K \to L$ is finite étale, we see that the trace pairing is nondegenerate (Lemma \refdiscriminant-lemma-discriminant) and hence there exists a $x \in L$ such that $\varphi_K(y) = \mathrm{Trace}_{L/K}(xy)$ for all $y \in L$. Then $x \in \mathcal{L}_{B/A}$ maps to $\varphi$ in $\omega_{B/A}$.

**Option E**

Recall that $\omega_{B/A} = \mathrm{Hom}_A(B, A)$ as $A \to B$ is finite. We send $x \in \mathcal{L}_{B/A}$ to the map $b \mapsto \mathrm{Trace}_{L/K}(bx)$. Conversely, given an $A$-linear map $\varphi : B \to A$ we obtain a $K$-linear map $\varphi_K : L \to K$. Since $K \to L$ is finite étale, we see that the trace pairing is degenerate (Lemma \refdiscriminant-lemma-discriminant) and hence there exists a $x \in L$ such that $\varphi_K(y) = \mathrm{Trace}_{L/K}(xy)$ for all $y \in L$. Then $x \in \mathcal{L}_{B/A}$ maps to $\varphi$ in $\omega_{B/A}$.

**Option F**

Recall that $\omega_{B/A} = \mathrm{Hom}_A(B, A)$ as $A \to B$ is finite. We send $x \in \mathcal{L}_{B/A}$ to the map $b \mapsto \mathrm{Trace}_{L/K}(bx)$. Conversely, given an $A$-linear map $\varphi : B \to A$ we obtain a $K$-linear map $\varphi_K : L \to K$. Since $K \to L$ is finite étale, we see that the trace pairing is nondegenerate (Lemma \refdiscriminant-lemma-discriminant) and hence there exists a $x \in L$ such that $\varphi_K(y) = \mathrm{Trace}_{L/K}(xy)$ for all $y \in L$. Then $x \in B$ maps to $\varphi$ in $\omega_{B/A}$.

**Option G**

Recall that $\omega_{B/A} = \mathrm{Hom}_A(B, A)$ as $A \to B$ is finite. We send $x \in \mathcal{L}_{B/A}$ to the map $b \mapsto \mathrm{Trace}_{L/K}(bx)$. Conversely, given an $A$-linear map $\varphi : B \to A$ we obtain a $K$-linear map $\varphi_K : L \to K$. Since $K \to L$ is finite étale, we see that the trace pairing is nondegenerate (Lemma \refdiscriminant-lemma-discriminant) and hence there exists a $x \in L$ such that $\varphi_K(y) = \mathrm{Norm}_{L/K}(xy)$ for all $y \in L$. Then $x \in \mathcal{L}_{B/A}$ maps to $\varphi$ in $\omega_{B/A}$.

> **Option H**
>
> Recall that $\omega_{B/A} = \text{Hom}_A(B, A)$ as $A \to B$ is finite. We send $x \in \mathcal{L}_{B/A}$ to the map $b \mapsto \text{Trace}_{L/K}(bx)$. Conversely, given an $A$-linear map $\varphi : B \to A$ we obtain a $K$-linear map $\varphi_K : L \to K$. Since $K \to L$ is finite étale, we see that the trace pairing is nondegenerate (Lemma \refdiscriminant-lemma-discriminant) and hence there exists a $x \in L$ such that $\varphi_K(y) = \text{Norm}_{L/K}(xy)$ for all $y \in L$. Then $x \in \mathcal{L}_{B/A}$ maps to $\varphi$ in $\omega_{B/A}$.

## C.2 Response from Qwen2.5-Instruct-72B

In this section we present the response from Qwen2.5-Instruct-72B below:

> **Response from Qwen2.5-Instruct-72B**
>
> To determine the correct proof, let's carefully analyze each step and identify any inconsistencies or errors.
> 1. **Definition of $\omega_{B/A}$**: All proofs correctly state that $\omega_{B/A} = \text{Hom}_A(B, A)$ when $A \to B$ is finite.
> 2. **Mapping from $\mathcal{L}_{B/A}$ to $\omega_{B/A}$**: All proofs correctly define the map sending $x \in \mathcal{L}_{B/A}$ to the map $b \mapsto \text{Trace}_{L/K}(bx)$.
> 3. **Converse Mapping**: - **Proof A**: Incorrect because it states that $\varphi$ is $B$-linear, but it should be $A$-linear. - **Proof B**: Correctly states that $\varphi$ is $A$-linear, but it incorrectly restricts the domain of $\varphi_K$ to $B$ instead of $L$. - **Proof C**: Correctly states that $\varphi$ is $A$-linear and correctly uses the nondegeneracy of the trace pairing over $L$. - **Proof D**: Incorrect because it defines $\omega_{B/A}$ as $\text{Hom}_A(B, B)$ instead of $\text{Hom}_A(B, A)$. - **Proof E**: Incorrect because it states that the trace pairing is degenerate, which contradicts the lemma. - **Proof F**: Incorrect because it states that $x \in B$ instead of $x \in \mathcal{L}_{B/A}$. - **Proof G**: Incorrect because it uses the norm instead of the trace. - **Proof H**: Correctly states that $\varphi$ is $A$-linear and correctly uses the nondegeneracy of the trace pairing over $L$, but it uses the term "finite separable" instead of "finite étale."
> Given these analyses, the correct proof is:
> $\boxed{C}$

## C.3 A Brief Analysis

In this section we give a brief analysis of the response from Qwen2.5-Instruct-72B, demonstrating the drawbacks of multiple-choice question format.

As we can see, since all 8 options are adapted from the same original proof, they all varies from the original proof at different places. Therefore, often in one place of modification, only one option differs from the other seven while the other seven options are the same at that place (as illustrated in Figure 3). This Phenomenon is also reflected in the response of Qwen2.5-Instruct-72B: This relatively weak model doesn't need complicated mathematical reasoning to deduce the correct answer, it can instead compare between the eight options to get the correct answer through a non-mathematical way.

# D More Comparisons with Mainstream Mathematical Benchmarks

We provide in Figure 6 a performance comparison of AlgGeoTest with MMLU-Pro(MATH) or GSM8K across various LLMs, similar to those of Section 4.3 and Figure 5.

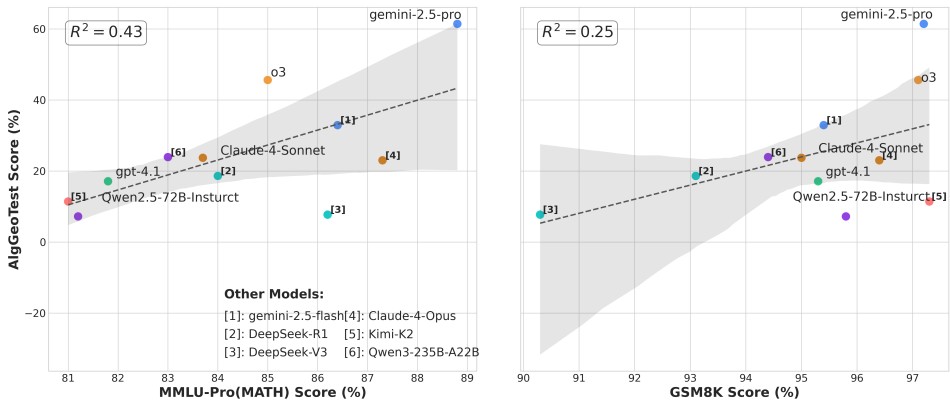

Figure 6: Scatter plot comparing AlgGeoTest and MMLU-Pro(MATH) or GSM8K. As in Figure 5, AlgGeoTest shows a weak correlation to MMLU-Pro(MATH) or GSM8K. Notice that the R-square value of AlgGeoTest versus GSM8K is comparatively low. This is probably because GSM8K is kind of out of date and thus the score of state-of-the-art LLMs on which is rather irregular.

