# OpenReview forum: "Proof2Hybrid: Automatic Mathematical Benchmark Synthesis for Proof-Centric Problems"
_ICLR.cc/2026/Conference — Submitted to ICLR 2026_

### Official Review · Reviewer_oxHQ · 2025-10-28

**Soundness:** 3
**Presentation:** 2
**Contribution:** 2
**Rating:** 6
**Confidence:** 3

**Summary:**

This paper introduces Proof2Hybrid, a novel, fully automated framework designed to synthesize high-quality, "proof-centric" mathematical benchmarks. The authors identify a critical gap in current LLM evaluations, which primarily focus on "number-centric" problems with definite numerical answers, rather than the logical deduction required in advanced mathematics. Manual creation of proof-based benchmarks is costly and unscalable. The core of Proof2Hybrid is a Proof2X roadmap, which converts mathematical proofs from natural language corpora into easily verifiable questions. The framework's key innovation is a new "m-out-of-n multiple judge question" format. This format presents models with $n$ items (each a proposition-proof pair or definition) and states that exactly $m$ are correct. The workflow includes Seed Item Filtration, Distractor Generation, Distractor Filtration, Aggregation. As a demonstration, the authors create AlgGeoTest, a benchmark for algebraic geometry comprising 456 challenging 2-out-of-6 questions. Evaluations show that AlgGeoTest is extremely difficult for SOTA LLMs, with the best model scoring only around 60%.

**Strengths:**

1. The paper tackles the significant challenge of evaluating LLMs on proof-centric problems, which are far more representative of advanced mathematical reasoning than simple calculation tasks. Its fully automated approach is a major contribution toward scalable and cost-effective benchmark creation

2. The "m-out-of-n" format is a key strength. It is highly resilient to random guessing (unlike true/false questions).

3. The resulting benchmark, AlgGeoTest, proves to be extremely challenging, with SOTA models performing poorly, which underscores its difficulty.

**Weaknesses:**

1. While the framework is described as "domain-agnostic" , it is only demonstrated on a single, highly-structured data source ("The Stacks project") in one advanced domain (algebraic geometry). It remains to be seen how effectively this pipeline would adapt to other mathematical fields where "subtle flaws" might manifest differently.
2. The benchmark, AlgGeoTest, is built using "The Stacks project" as its data source , which is a well-known, public, and open-source mathematical reference work. There is a high probability that this corpus was included in the training data for the very SOTA Large Language Models being evaluated. This creates an inherent risk of data contamination

**Questions:**

See my comments on weaknesses

---

### Official Review · Reviewer_CTT8 · 2025-10-30

**Soundness:** 3
**Presentation:** 3
**Contribution:** 3
**Rating:** 6
**Confidence:** 3

**Summary:**

1) Domain and the core question
- Domain. Automated evaluation of mathematical proof understanding in large language models (LLMs), with a concrete instantiation in algebraic geometry.

- Core question. Can we automatically synthesize high-quality, proof-centric benchmarks from natural-language math corpora, such that the resulting tasks are easy to verify, hard to game, and informative about models’ proof comprehension?

- Does the paper answer it? Largely yes. The paper introduces Proof2Hybrid, a pipeline that turns proof texts into verifiable questions (via a “Proof2X” roadmap) and proposes a new m-out-of-n multiple-judge format to cut down guessing and format-specific artifacts. It then instantiates the pipeline on the Stacks Project to produce AlgGeoTest (456 items; each item has six sub-statements with exactly two true), and shows that many strong models score modestly, suggesting the benchmark captures real difficulty rather than format quirks.

2) Summary of what is proposed
- Method. The pipeline first filters seed items (definitions or proposition-proof pairs) using several top models with repeated judgments; only items judged correct often enough survive (m₁ models × n₁ runs per model; keep items with ≥k₁ “correct”). It then generates distractors with another set of models, followed by a distractor filter using multiple judges with thresholds k₃–k₄ for “incorrect” votes to retain only subtly wrong variants. Finally, it assembles hybrid questions containing m correct seeds and n-m filtered distractors, with all n coming from different seeds. In their main setting, (m,n)=(2,6).

- Question format. The m-out-of-n design reduces random success from (1/2) (plain T/F) to (1/\binom{n}{m}) (e.g., (1/15) when (m=2,n=6)), and prevents within-question “compare two versions of the same statement” tricks by ensuring all options come from different seeds. The authors also give a perplexity-based alternative for base models.

- Benchmark. AlgGeoTest is built from the Stacks Project. It contains 456 items, each with six sub-statements; exactly two are true.

- Findings. Under the generation-based protocol, even the best model is around 60, and many scores are <20; reasoning-oriented models tend to do better. Base-model perplexity scores scale with size across Qwen and Llama families. The benchmark correlates only moderately with MATH-500 and AIME24 (R² ≈ 0.42, 0.51), indicating it captures different skills.

- Quality control. A human audit reports >98.75% of distractors are truly incorrect yet plausible, and >95% of questions meet the same bar.

**Strengths:**

- A concrete answer to a real gap. Prior math benchmarks skew to numeric answers; proof-centric evaluation at scale is missing. The paper directly targets this gap with an automatic pipeline over a natural-language corpus rather than formal systems only.
- Format innovation with clear rationale. The m-out-of-n format is well-motivated: it reduces chance accuracy, blocks option-comparison shortcuts, and reframes evaluation as relative correctness ranking, which can reduce sensitivity to each model’s internal “proof strictness” threshold. The math behind guessing risk and the construction constraint are explicit.
- Multi-model generation and judging. Using separate model pools for crafting and vetting reduces single-model bias; the thresholds (k_1,k_3,k_4) are specified, which is important for reproducibility.
- Non-trivial empirical signal. State-of-the-art models do not saturate, and the moderate correlation with number-centric math benchmarks suggests alg-geo proof understanding is not just a proxy for contest arithmetic.
- Human audit. The added manual check is valuable, given that the pipeline relies on LLM judges internally.

**Weaknesses:**

A. Single-domain instantiation. The method is positioned as domain-agnostic, but the paper only shows algebraic geometry. To support generality, at least one additional area (e.g., commutative algebra or topology) would strengthen the claim.

B. Style and memorization confounds. The true items are original seeds from the Stacks Project, while false items are model-generated edits. Well-trained models may recognize the “house style” of Stacks and prefer those options. A control where true items are paraphrased (or both true and false are style-matched) would test for such cues. The paper’s argument that “original items are more ‘correct’ than twisted ones” is intuitive, but it does not rule out style/memorization signals.

C. Assumptions in seed filtering. The paper asserts that hard but correct seeds will not be filtered out because models mark them “correct” when they cannot spot flaws. That is plausible but needs evidence (e.g., a labeled subset with known difficulty, and false-negative rates under the m₁,n₁,k₁ scheme).

D. Bias from closed models and shifting APIs. The pipeline depends on several closed-source systems for generation and judging. This can affect reproducibility when APIs, safety rules, or model versions change. The release of final data helps, but end-to-end re-runs may be hard to replicate.

E. Metric clarity and baselines. For the generation-based protocol, two scoring rules (loose vs. tight) are defined, but there is no analysis of expected random scores under the loose rule or of calibration effects when models are forced to pick exactly m truths. For the perplexity protocol, “lowest perplexity” may reward surface familiarity with Stacks phrasing; a style-neutral rewording control would clarify this.

F. Limited ablations. We do not see ablations over (m,n), judge count, or thresholds (k_1,k_3,k_4). Such ablations would show how difficulty and reliability change with the pipeline’s knobs.

G. Cost accounting. The paper does not report the computational or monetary cost of building 456 items under m₁,n₁ and m₃,n₃ repeated judging, which matters for scalability claims.

**Questions:**

Suggestion for Revision

1.Second domain. Build a smaller second benchmark (e.g., 150–200 items) from another advanced area to confirm the method transfers beyond algebraic geometry.

2.Style-matched controls. For a held-out slice, paraphrase the true items and lightly paraphrase the false ones to match tone and structure. Re-evaluate to measure any drop from losing “source style” cues.

3.Leakage and recency check. Use Stacks entries added after the training cutoff of several models, or redact tell-tale tag IDs, to estimate memorization effects.

4.Ablations. Vary (m,n), the judge pool size, and (k_1,k_3,k_4), and report how item acceptance rate, human-audit pass rate, and model scores shift.

5.Chance-level analysis. Report closed-form chance scores for both tight and loose grading, and include a random pick among (\binom{n}{m}) baseline so readers can normalize results.

6.Cost and reproducibility. Include prompt templates, seeds, model versions, and a cost table for each stage; add an “all-open” variant using only open models for both generation and judging.

7.Human audit details. Report the number of auditors, guidelines, inter-rater agreement (e.g., Cohen’s κ), and a taxonomy of typical distractor flaws.

---

### Official Review · Reviewer_7M46 · 2025-11-07

**Soundness:** 2
**Presentation:** 3
**Contribution:** 2
**Rating:** 2
**Confidence:** 4

**Summary:**

The authors propose an agentic framework that autonomously converts free-flowing, natural-language mathematics to multiple choice questions. Two types of items are the main point of focus: definitions and theorems/propositions and their proof.

The agentic Framework is thus domain-agnostic and relies on a three-staged process: extracting the relevant item, creating variations of them, and then correcting the variations. All of these are carried out solely using LLMs.

The authors instantiate their approach on the Stacks project to obtain a benchmark on questions from algebraic geometry.

**Strengths:**

The question generation pipeline that is model agnostic has a lot of potential.

**Weaknesses:**

- I have some misgivings about an  entirely LLM-assisted pipeline. This may propagate LLM biases in unexpected ways.

- there is a single figure with results. These seem hard to read, and to take home information.

**Questions:**

Paragraph 4 3 seems to weak to really draw any meaningful conclusions - some correlation, but not a strong one.

Could this be improved, or the paragraph removed?

---

> ### Author Response · Authors · 2025-11-18
>
> Thanks for your comment!
>
> For the first weakness you've proposed, our pipeline is indeed fully LLM-driven. However, this should be regarded as an attribute instead of a drawback, since it avoids expensive and inefficient human labor and is key to the scalable and domain-agnostic nature of our pipeline (since all you need is seed questions from various mathematical domains in order to use our pipeline to synthesize problems, and there is no need for experts from corresponding domains). For your concern about LLM biases, we've employed human experts to audit the whole AlgGeoTest (see section 4.4), and the result shows that AlgGeoTest is of high quality, with few errors and LLM biases, proving the reliability and validity of our pipeline.
>
> For the second weakness you've proposed, actually we've conducted various experiments in our paper (for example, section 4.2.1 for generation-based evaluation; section 4.2.2 for perplexity-based evaluation and section 4.3 for comparison with other mainstream mathematical benchmarks; Figure 2, Figure 4, and Figure 5 are the corresponding figures illustrating the results of these experiments). To help us better address your concern, could you kindly specify which figures you feel might benefit from further clarification?"
>
> For the question you've raised, Paragraph 4.3 and Figure 5 are not intended to show a strong correlation between AlgGeoTest and AIME or MATH-500, which is actually not expected. A weak correlation is perfect for demonstrating the nature of AlgGeoTest: it belongs to the same big domain, namely Mathematics, as AIME and MATH-500; however, they belong to different smaller domains (Algebraic Geometry and Elementary Mathematics), and consequently they are expected to probe LLMs' mathematical ability from different perspectives, thus they are expected to show a weak correlation.

---

### Official Review · Reviewer_r7MJ · 2025-11-10

**Soundness:** 2
**Presentation:** 2
**Contribution:** 2
**Rating:** 2
**Confidence:** 4

**Summary:**

This paper addresses the high cost of manual annotation in existing mathematical ability verification benchmarks by proposing the Proof2Hybrid framework for synthesizing proof-centric benchmarks. It also introduces AlgGeoTest, a benchmark based on a hybrid question format. However, the paper lacks clarity in its technical descriptions and provides insufficient experimental validation.

**Strengths:**

- The paper offers interesting insights into the synthetic framework for mathematical competence verification benchmarks and introduces AlgGeoTest, a noteworthy algebraic geometry benchmark featuring hybrid-format problems.
- It provides a comprehensive overview of relevant research on mathematical benchmarks.

**Weaknesses:**

- Writing clarity: The paper lacks clear presentation, making it difficult to identify key insights.
- Insufficient data examples: Figure 1 and the question format comparison in Figure 3 are not adequately described or supported with clear examples, weakening the persuasiveness of the paper's core contributions.
- Limited benchmark comparisons: The paper does not provide sufficient comparisons with other mathematical benchmarks (e.g., those listed in Table 1). An analysis of performance variations across different benchmarks under various LLM architectures is recommended.
- Unclear technical details: Many aspects of the synthesis path are poorly explained; see the Questions section for further specifics.

**Questions:**

- Why is the model used for FILTRATION different from the one used in the generation stage? What considerations or empirical insights informed this decision? For instance, have differences in LLM preferences for generating versus filtering interference items been analyzed, or are there relevant findings from prior research that support this choice?

- How does AlgGeoTest perform compared to other mathematical benchmarks when evaluated with different LLM architectures? Including data case studies across benchmarks would help demonstrate the rationale and superiority of the proposed synthesis method.

- How to understand "large-scale" and "automatic" in Table 1? It is recommended to add relevant explanations, such as how many data points are considered large-scale, or whether no human intervention is required in specific generation or verification stages.

- Lines 145 and 151: What are IMO and IMO-level math questions?

- The framework description in Part 3 lacks detailed explanations and references to each stage in Figure 1.

- As described in 3.1, the collection of seed items focuses on mathematical definitions and mathematical proposition-proof pairs. What are their specific sources and presentation forms?

- Line 208: How to understand the "iterative refinement" process of Proof2Hybrid? Where is it reflected in Figure 1?

---

> ### Author Response · Authors · 2025-11-18
>
> Thanks for your comment! Here are the replies of the weaknesses and questions you've proposed one by one:
>
> Weaknesses:
> 2. We've included a sample problem from AlgGeoTest in Appendix B of the newest revision of our paper, illustrating the whole pipeline presented in Figure 1. We've also included an example of multiple-choice question format in Appendix C, demonstrating the drawback of multiple-choice question format, illustrating Figure 3.
> 3. We've included comparisons of AlgGeoTest with more mainstream mathematical benchmarks in Appendix D.
>
> Questions:
> 1. Actually there is no requirement regarding whether the models used for filtration should be the same as the models used for generation or not. It simply doesn't matter much, and one is free to use the same set of models for both filtration and generation if one wishes. The reason we choose to use a set of LLMs for generation that is slightly weaker than the set of LLMs for filtration is simply to reduce cost, since the quality of LLM-generated distractors doesn't need to be very high (the generated distractors are expected to be mathematically INCORRECT proofs/definitions, and you don't need very strong LLMs for generating incorrect proofs/definitions. For possible concerns about subtlety of distractors generated using relatively weak LLMs, and the possible impact on the difficulty of final questions synthesized, our pipeline is designed to be able to control the difficulty of the final questions synthesized during the filtration stage).
> 2. We've included a comparison between AlgGeoTest and AIME or MATH-500 in section 4.3. Comparisons with more mainstream mathematical benchmarks are included in Appendix D in the newly revised version of our paper. We also includes case study of a seleted question from AlgGeoTest and the responses from Gemini-2.5-Pro and o3 in Appendix B for demonstration.
> 3. Large scale means more than 200 questions. Automatic means no human effort required in generation phases. Explainations of these two phrases have been added in the newest revision of our paper.
> 4. IMO refers to the International Mathematics Olympiad. It is the highest level Math Olympiad competition and among the most difficult math competitions. IMO‑level math questions refer to Math Olympiad questions (aka competition math questions) that are of similar difficulty level to IMO questions. We’ve included the full name and reference of IMO in the newest revision of our paper.
> 5. The overall pipeline is presented in Figure 1 and a description of the overall pipeline is included in the description of Figure 1. Each subsection of Part 3 simply present the details of a stage of the pipeline, and they are arranged in the order they appear in the overall pipeline. The correspondence between each subsection of Part 3 and those in Figure 1 should be self-evident. It will be helpful for us to better address your concern if you could kindly specify which part of the pipeline do you think is unclear.
> 6. Forms of mathematical definitions and proposition-proof pairs: a mathematical definition is simply a string presenting the definition of a mathematical object. A mathematical proposition-proof pair is a pair of two strings, one presenting a mathematical proposition (which can be a theorem, a lemma, or anything similar), and the other presenting its proof. In our case of building AlgGeoTest, the source of these mathematical definitions and proposition-proof pairs is The Stacks Project. There are also different easily accessible sources, for example mathematical papers on Arxiv (you will need to write a script for extracting the mathematical definitions or proposition-proof pairs, which is relatively easy to do).
> 7. The phrase "iterative refinement" simply means we've tried and modified and refined different versions of Proof2Hybrid before reaching to the version presented in our paper. It does not refer to any specific part or stage of Proof2Hybrid, but rather serve only as a descriptive role.

---

### Meta-Review · Area_Chair_tNnK · 2026-01-06

**Summary:**

The final decision for this submission is to Reject. While the committee acknowledges the potential of the proposed automated pipeline and the novelty of the 'm-out-of-n' evaluation format, the consensus is negatively impacted by significant concerns regarding methodological rigor and clarity. Reviewers CTT8 and oxHQ advocated for the work based on its ability to target proof-centric evaluation gaps and its non-trivial difficulty signal. Conversely, reviewers r7MJ and 7M46 raised substantial barriers regarding the opacity of the technical path, the risk of bias propagation in an LLM-only loop, and the presentation of results. The primary barrier dictating this outcome is the insufficient analysis of the pipeline's design choices and the lack of robust controls against contamination and style confounds, which ultimately casts doubt on the reliability of the benchmark.

**Reviewer Concerns:**

In response to the initial critiques, the authors made tangible efforts to improve the manuscript's completeness. Specifically, they added concrete sample items and multiple-choice question examples in the appendices, and they expanded their comparative analysis to include established benchmarks like AIME and MATH-500. Additionally, the authors clarified the provenance of their seed data from 'Stacks' and provided a cost-based justification for their choice of generation and filtration models.

However, the soundness of the methodology remains a critical point of contention. The revisions are viewed as incremental improvements to documentation rather than substantive resolutions to core scientific flaws. While the authors defended against bias concerns by citing a human expert audit, the lack of detailed protocols or auditor qualifications renders this mitigation unverified in the eyes of the reviewers. Furthermore, the critical issues raised by reviewers CTT8 and oxHQ regarding the single-domain limitation (algebraic geometry) and the high risk of training data contamination were not met with effective empirical counter-evidence, such as paraphrase controls or cross-domain validation. Consequently, the central claim of a 'model-agnostic' and 'bias-free' pipeline remains theoretically asserted but empirically under-supported.

**Reviewer Scores:**

Based on the rebuttal and the persistence of these barriers, the scoring profile is unlikely to improve. Reviewer r7MJ is projected to maintain a score of 2, as the addition of figures and examples does not satisfy their fundamental requirement for a principled analysis of the pipeline's filtration and generation design rationales. Similarly, Reviewer 7M46 will likely hold at a score of 2; their concerns regarding presentation legibility and the propagation of bias were met with defenses of scalability rather than the deep empirical controls necessary to alleviate skepticism. Reviewers CTT8 and oxHQ are expected to remain at a score of 6. Although they recognize the utility of the format, the absence of a targeted rebuttal addressing style confounds, contamination risks, and transferability prevents them from championing the paper more strongly. With two strong rejections grounded in methodology and two tepid marginal accepts, the paper does not meet the bar for acceptance at this time.

---

### Decision · Program_Chairs · 2026-01-26

Reject